# Compatibility Investigation of a Steroid and Two Antibiotics with Heparin for the Prevention of Catheter Occlusion in Neonatal Intensive Care Units

**DOI:** 10.3390/mps8060136

**Published:** 2025-11-06

**Authors:** Mao Maekawa, Masamitsu Maekawa, Yu Sato, Shimpei Watanabe, Masatoshi Saito, Nariyasu Mano

**Affiliations:** 1Department of Pharmaceutical Sciences, Tohoku University Hospital, Sendai 980-8574, Japan; mao.maekawa.d3@tohoku.ac.jp (M.M.); nariyasu.mano.c8@tohoku.ac.jp (N.M.); 2Faculty of Pharmaceutical Sciences, Tohoku Medical and Pharmaceutical University, Sendai 983-8536, Japan; sato.yu@tohoku-mpu.ac.jp; 3Center for Perinatal and Neonatal Medicine, Tohoku University Hospital, Sendai 980-8572, Japan; shimpei.watanabe.c2@tohoku.ac.jp (S.W.); masatoshi.saito.b4@tohoku.ac.jp (M.S.)

**Keywords:** neonatal intensive care units, simultaneous infusion, stability, co-administration, compatibility

## Abstract

Intravenous medications are frequently administered through shared catheter lines in neonatal intensive care units (NICUs) due to the limited venous access in preterm infants, raising concerns about drug incompatibilities that may cause serious complications. Hydrocortisone sodium (HDC), ampicillin (ABPC), and cefotaxime (CTX) are commonly used in NICUs and are often co-administered with unfractionated heparin (UFH), which is routinely infused to prevent catheter occlusion. This study evaluated the physicochemical compatibility of HDC, ABPC, and CTX when mixed with UFH. Each drug was combined with UFH at equal volumes, and the mixtures were assessed immediately and after 3 h of storage by visual inspection, pH measurement, UV absorbance, and HPLC-UV analysis. No precipitation, turbidity, or color changes were observed in any mixture, and UV absorbance showed no relevant deviations compared with controls. Slight pH variations were detected but remained within acceptable limits. In semi-quantitative HPLC analysis, relative peak area changes were all below 10%, indicating no major degradation of the drugs. These findings suggest that HDC, ABPC, and CTX maintain acceptable physicochemical compatibility when co-administered with UFH, supporting their safe concomitant use in NICU practice.

## 1. Introduction

Various injections of nutrition and sedatives are frequently administered in neonatal intensive care units (NICUs). A study on intravenously administered drugs in the NICU found that the primary categories included anti-infective agents, cardiovascular drugs, and sedatives [1]. The method of administration varies depending on the specific medication used, with both intermittent and continuous infusions being used.

In some cases in routine NICU practice, the number of drug infusions exceeded the available intravenous (IV) lines. In such cases, the options are to either secure an additional IV line, which carries risks such as mechanical injury, infections, and thrombosis, or administer multiple drugs through a single line via Y-site administration. For the Y-site administration, each drug was prepared separately and mixed with the Y-site connector before entering the bloodstream [2]. However, when drugs are mixed within a catheter, precipitation may occur because of differences in their physicochemical properties [3,4]. Furthermore, there have been reports of fatal embolisms caused by the injection of incompatible drugs [5,6].

Concerning the compatibility of injections in the NICU clinical practice, reviews have utilised databases [2,4,7]. However, because of the limited availability of information, investigating the compatibility of unverified combinations remains highly meaningful.

Hydrocortisone sodium succinate (HDC) is used in the NICUs to treat adrenal insufficiency and bronchopulmonary dysplasia [8,9,10,11,12]. However, the HDC product information indicates that it has a pH-dependent compatibility concern, and it has been shown to produce a white precipitate under acidic conditions and a yellow precipitate under alkaline conditions [13]. Therefore, caution should be exercised when HDC are administered simultaneously with other drugs.

Complications associated with peripheral percutaneous central venous catheters in neonates include occlusion. Unfractionated heparin (UFH) is occasionally administered to prevent these complications [14,15,16]. Owing to its short half-life, UFH requires continuous administration. However, there are concerns regarding the compatibility with other drugs administered simultaneously through the Y-site of the catheter.

Previous studies examined the compatibility of UFH [17,18,19,20] and HDC [21,22,23] with other drugs. In particular, the combination of UFH and HDC is considered incompatible based on the UFH product information [24], because crystal precipitation occurs after 3 h when UFH and HDC are mixed. However, this test used undiluted UFH (1000 U/mL), and the conditions were different from those used in NICU clinical practice when using low-concentration UFH to prevent catheter blockage. Therefore, it is clinically important to elucidate the compatibility between UFH and HDC in NICU settings. In the NICU, UFH is typically diluted to at most 20 U/mL to meet neonatal dosing requirements (maximum 50 IU/kg/day in infants weighing 0.5–5 kg) and is administered at infusion rates of 0.5–1 mL/h [14,15,16], indicating the need to evaluate compatibility under these clinically relevant conditions.

Sepsis is a major cause of neonatal mortality; therefore, antibiotics are frequently administered in NICUs [25,26,27]. Among these, ampicillin sodium (ABPC) and cefotaxime sodium (CTX) are frequently used as empirical treatments for early-onset neonatal sepsis [28,29,30,31]. The potential incompatibility of ABPC [32,33,34] and CTX [35,36,37] with other drugs has been previously reported. Although some of the physicochemical properties have been verified, the available information remains limited. Baker et al. [37] suggested that the content of a mixture of UFH and CTX in an antibiotic locking solution may decrease. However, no clear conclusions have been drawn regarding the optimal conditions for UFH mixing to prevent catheter occlusion.

This study aimed to evaluate the physical and chemical compatibility of HDC, ABPC, and CTX—three drugs frequently co-administered in NICUs—when in contact with 20 U/mL UFH under conditions simulating Y-site administration. The concentrations of each drug were set at their maximum clinically used concentrations. Because these antibiotics can be administered intermittently to avoid direct admixture, compatibility among HDC, ABPC, and CTX themselves was not investigated.

In general, the evaluation of admixture stability requires the consideration of both physical and chemical compatibility. For physical compatibility, the primary assessment involves visual inspection of any changes in appearance such as precipitation, turbidity, or discoloration. Turbidity [21,23] and absorbance measurements [18,20,38] are often used for quantitative evaluation. As pH can affect drug stability and solubility, pH variation before and after mixing is also an important parameter [18,20,38,39].

High-performance liquid chromatography (HPLC) is commonly used to quantify degradation or transformation of active ingredients [39]. HPLC-UV analysis is widely used to evaluate the degradation of low-molecular-weight drugs. In contrast, unfractionated heparin (UFH) is a highly sulfated polysaccharide with a broad molecular weight distribution ranging from approximately 5000 to 20,000 Da [24], making it unsuitable for separation or quantification by conventional HPLC-UV methods. Therefore, in this study, quantitative analysis was limited to the three low-molecular-weight compounds, whereas UFH compatibility was assessed primarily based on physical parameters such as turbidity, precipitation, and pH change.

Therefore, using this combined analytical approach, we investigated the compatibility of HDC, ABPC, and CTX with UFH under conditions simulating continuous infusion and Y-site administration in NICU clinical practice.

## 2. Materials and Methods

### 2.1. Drugs and Reagents

The drugs used in the NICU clinical practice at Tohoku University Hospital were investigated, and the concentrations were set at the maximum concentration used in the clinical setting. The concentration of UFH was set to 20 U/mL, considering the maximum dose of 50 IU/kg/day [14,15,16], based on neonatal weight (0.5–5 kg) and infusion flow rate (0.5–1 mL/h). Additionally, ABPC and CTX, which can be administered as boluses, were prepared at 250 mg/mL in saline, which was the maximum solubility concentration confirmed for each formulation. HDC was dissolved in water for injection provided with the formulation and diluted with saline to a final concentration of 50 mg/mL (Table 1).

Sodium phosphate corrective injection 0.5 mmol/mL (Otsuka Pharmaceutical Factory, Inc., Tokushima, Japan) and an 8.5% calcium gluconate injection (Nichi-Iko Pharmaceutical Co., Ltd., Toyama, Japan)) (Ctrl [+]) were used as positive controls.

The mobile phase consisted of formic acid (Nacalai Tesque, Kyoto, Japan; ≥98.0% (T), Specially Prepared Reagent), ultrapure water (Organo Corporation, Tokyo, Japan), and acetonitrile (Kanto Chemical Co., Inc., Tokyo, Japan; for HPLC, >99.9% (GC)).

### 2.2. Mixing of Drugs for Preparation of Test Samples

The UFH solution was mixed with the HDC, ABPC, and CTX solutions in a ratio of 1:1 at room temperature under fluorescent lighting at an average of 199 lx. The resulting mixtures were prepared as follows: Mix A (10 U/mL UFH + 25 mg/mL HDC), Mix B (10 U/mL UFH + 125 mg/mL ABPC), and Mix C (10 U/mL UFH + 125 mg/mL CTX). After mixing, the required amount of each mixture was removed immediately and again after 3 h and used for various tests (Table 2).

As a positive control for appearance changes (Ctrl [+]), a solution containing equal amounts of sodium phosphate corrective injection and calcium gluconate injection was prepared. Water was used as a negative control (Ctrl [−]). As controls for the chemical changes, 25 mg/mL HDC, 125 mg/mL ABPC, and 125 mg/mL CTX were prepared in the absence of UFH.

### 2.3. Visual Inspection

All solutions were sampled once and visually observed for crystal precipitation, suspension, and colour change under black and white backgrounds, and photographs were taken at room brightness, as described in Section 2.2.

### 2.4. Absorbance Measurement

The ultraviolet (UV) absorbances of the solutions were measured using an Infinite M200 spectrophotometre (TECAN, Männedorf, Switzerland). Prior to the experiment, the absorbance of water with the lowest absorbance at 685 nm was used for evaluation.

Normal saline (0.9% NaCl injection), compliant with JP <6.06> and <6.07>, was used as the negative control and blank for spectrophotometric analysis. Mixtures exhibiting absorbance comparable to saline were considered free of precipitation or turbidity.

All solutions were sampled in triplicate, measured at the initial and 3 h time points. The average values were used for analysis. Crystal precipitation and suspension in the mixtures were statistically assessed using the Student’s *t*-test to compare differences with the negative control.

### 2.5. pH Measurement

The pH of the mixture was measured three times using a LAQUA F-71 (HORIBA, Kyoto, Japan) on a single sample, at the initial and 3 h time points. The change in the average value was evaluated [20]. The pH changes in the mixtures were statistically assessed using Student’s *t*-test.

### 2.6. Measurement of Changes in HDC, ABPC, and CTX Content

HPLC/UV was employed to evaluate the changes in the HDC, ABPC, and CTX contents in each mixture. The HPLC system used was a NexeraX2 (SHIMADZU, Kyoto, Japan), and UV analysis was performed using an SPD-M20A (Prominence, SHIMADZU, Kyoto, Japan). For separation, the InertSustain C18 PEEK (2.1 mm i.d. × 150 mm, 2 μm, GL Sciences, Tokyo, Japan) was used, with a mobile phase A (formic acid/water [0.1:100, *v*/*v*]) and mobile phase B (formic acid/acetonitrile, [0.1:100, *v*/*v*]), respectively. The gradient program was: 0.00–1.00 min, 20% B; 1.00–2.00 min, 20%→40% B; 2.00–5.00 min, 100% B; post-run, re-equilibration to 20% B (Table 3). The total chromatographic run time was approximately 5 min. Detection wavelengths were set at 254 nm for HDC and CTX and 230 nm for ABPC. All solutions were sampled in triplicate and analysed at the initial and 3 h time points. The average peak areas were integrated and the percentage change (%) was determined relative to a solution of equivalent concentrations (25 mg/mL HDC, 125 mg/mL ABPC, and 125 mg/mL CTX) without UFH.

## 3. Results

### 3.1. Change in Appearance with Visual Inspection

At the initial and 3 h time points, the mixed solutions of Mixes A and B were clear and colourless, whereas Mix C was clear and yellow, with no precipitation or suspension of crystals (Figure 1a–f,i–n). In contrast, the positive control sample became turbid at the initial point (Figure 1g,h).

### 3.2. Absorbance

The UV absorbance of each mixture, both at the initial and 3 h time points, showed no significant difference compared with the saline control (Table 4). This indicates that no significant spectral changes occurred, suggesting the absence of visible or subvisible particulate formation.

### 3.3. pH Change

The pH of Mixes A and C increased after 3 h, whereas that of Mix B decreased (Table 5).

### 3.4. Changes in HDC, ABPC, and CTX Content

The changes in the HPLC/UV peak areas for HDC, ABPC, and CTX were all within 10% of the control, both at the initial and 3 h points (Table 6). Representative chromatograms of the mixtures at the initial and 3 h points are shown in Figure 2.

Each drug exhibited a single, well-defined peak at the expected retention time without interference from coexisting compounds, indicating sufficient selectivity of the analytical method.

No degradation peaks or new signals were detected within the observed retention range (0.5–3.0 min).

## 4. Discussion

UFH is frequently performed to prevent catheter occlusion [14,15,16]. Owing to its short half-life, continuous administration is required. When venous access is limited, other medications are often coadministered through the same catheter. Although HDC and antibiotics are widely used in NICUs, their compatibility with UFH has not yet been fully elucidated.

In this study, we investigated whether the compatibility of HDC, ABPC, and CTX with UFH to prevent catheter occlusion could be sufficiently addressed by assessing the changes at the initial and 3 h time points, thereby eliminating the need for further verification. Generally, these drugs are administered in boluses and prepared separately from UFH. Although continuous administration of HDC has been compared to bolus administration in terms of efficacy and safety [40,41], the superiority of continuous administration has not been established. Consequently, this study did not consider real-world scenarios in which heparin and HDC are directly mixed in the same container. A typical peripheral central venous catheter features an internal diameter ranging from 0.012 to 0.032 in (approximately 0.305–0.813 mm) [42], a length of 60 cm, and an internal volume of approximately 0.311 mL. At a slow infusion rate of 0.5 mL/h via a syringe pump [43], the drugs within the catheter are expected to mix in less than 1 h. Consequently, a three-hour period was considered sufficient to evaluate the compatibility of the drugs within the catheter.

For example, in Kenneally et al. [18] and Dobson et al. [20], absorbance measurements were used along with visual inspection to assess physical stability. In the present study, we applied this approach and measured the absorbance at 685 nm; water exhibited the lowest absorbance. For visual inspection, a mixture containing a phosphate preparation (sodium hydrogen phosphate hydrate and sodium dihydrogen phosphate hydrate) and calcium gluconate, which is known to form poorly soluble calcium phosphate salts, was used as a positive control [44]. Visual inspection confirmed the absence of precipitation in any of the mixtures (Figure 1). Furthermore, the absorbance measurements revealed no changes in the mixtures, with values comparable to those of the negative control (Table 4). These results quantitatively demonstrated that UFH and the respective drugs in each mixture did not form insoluble particulate matter within 3 h of mixing.

According to the product information for HDC, the pH after dissolution typically ranges from 7.0–8.0. HDC formed white and yellow precipitates under acidic and alkaline conditions, respectively [13]. In this study, the pH of HDC increased by +0.063 after mixing. However, since HDC is known to remain stable within the pH range of 7.0–8.0, this slight increase is within the range that does not affect its stability. This stability was likely due to the buffering action of anhydrous sodium monohydrogen phosphate and anhydrous sodium dihydrogen phosphate, which were included as additives in the preparation. These buffering agents appeared to minimise the pH fluctuations, thereby preventing the formation of precipitates in the mixed solutions.

According to a report by Andrew [22], mixing HDC with midazolam or ciprofloxacin results in a content change exceeding 10%. In contrast, this study found that the change in the HDC content when mixed with UFH was within 10% of that of the unmixed HDC solution (Table 6). This suggests that there may be no issues with chemical compatibility. While the product information for UFH indicated that crystals formed after 3 h when UFH and HDC were mixed at their original concentrations, it is considered acceptable to administer HDC simultaneously with UFH at low concentrations, such as 20 U/mL.

The product information indicated that the potency of ABPC decreased under both acidic and alkaline conditions when the pH was outside the stability range. Previous reports [33,34] have documented cases of physical instability when ABPC was mixed with other drugs. In this study, the compatibility of ABPC with the UFH was evaluated. While a time-dependent decrease in pH and potency was observed, the stability criteria were maintained within a 3 h timeframe. However, prolonged mixing increases the likelihood of compromising stability.

For CTX, there are concerns regarding potency reduction, precipitation, and discoloration when mixed with other drugs, as noted in the product information. Baker [37] reported the chemical and physical instabilities of a mixture of UFH and CTX in an antibiotic lock solution. In this study, the solution was yellow and clear at the initial point, reflecting the original colour of the product, with no changes observed in its properties or absorbance, even after 3 h. The pH testing and content analysis also met the product standard range, suggesting that the physicochemical stability of the mixture may be acceptable.

The pH of Mix B deviated from the specified range for heparin (pH 5.5–8.0). Heparin is a mucopolysaccharide, and considering its structural characteristics, it is unlikely to undergo a phase transition from an ionic to a solid state under the pH conditions observed in this study. Furthermore, in previous studies [19], a mixture of 10 mg/mL ABPC and 10 U/mL heparin was prepared and tested over 14 days, with no clinically significant changes in APTT. Based on these findings, the pH fluctuations observed in this study are unlikely to affect the physicochemical properties of heparin.

In studies evaluating the compatibility of injectable admixtures, physicochemical assessments such as visual inspection, turbidity, and absorbance measurements are often used, whereas evaluation of chemical stability is frequently limited to pH measurements. In this study, changes in the content of low-molecular-weight compounds were detected using HPLC-UV analysis. The peak areas of HDC, ABPC, and CTX in the mixed solutions were compared with those in solutions without UFH to assess content changes at the initial and 3 h time points.

This study also evaluated changes in the appearance, pH, and content of HDC, ABPC, and CTX in solutions mixed with UFH. However, one limitation is that it did not assess possible content or structural changes in UFH, polymerized compounds, or other potential reaction products. Another limitation is the absence of calibration curves and analytical method validation data for the HPLC-UV measurements. Because calibration curves were not constructed and the analytical method was not formally validated for linearity, precision, or accuracy, quantitative comparison of peak areas could be affected by detector noise or matrix effects. Therefore, the results should be interpreted as semi-quantitative, reflecting relative rather than absolute changes in analyte concentrations.

## 5. Conclusions

These results suggest that the compatibility of HDC, ABPC, and CTX with UFH in the catheter mixture to prevent catheter obstruction was acceptable. As there are limited intravenous routes available for the intensive care of preterm infants, it is clinically important to investigate the compatibility between drugs administered simultaneously through the same catheter. Applying the methods used in this study to examine the compatibility of drugs that have not yet been verified will lead to the development of safe and effective drug treatments. Based on the actual use of drugs in our hospital’s NICU, we plan to further investigate the compatibility of additional drug combinations in the future.

## Figures and Tables

**Figure 1 mps-08-00136-f001:**
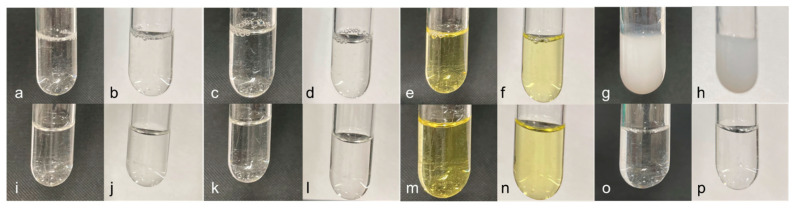
Change in appearance of the mixture. (**a**) Mix A: initial, black background, (**b**) Mix A: initial, white background, (**c**) Mix B: initial, black background, (**d**) Mix B: initial, white background, (**e**) Mix C: initial, black background, (**f**) Mix C: initial, white background, (**g**) Ctrl (+): black background, (**h**) Ctrl. (+): white background, (**i**) Mix. A: after 3 h, black background, (**j**) Mix. A: after 3 h, white background, (**k**) Mix B: after 3 h, black background, (**l**) Mix B: after 3 h, white background, (**m**) Mix C: after 3 h, black background, (**n**) Mix C: after 3 h, white background, (**o**) Ctrl (−): black background, (**p**) Ctrl (−): white background.

**Figure 2 mps-08-00136-f002:**
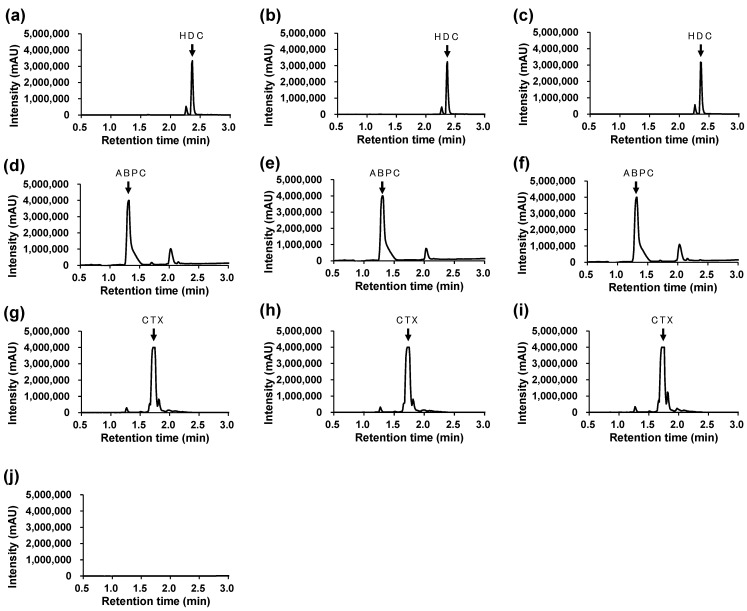
Representative HPLC/UV chromatograms. (**a**) Ctrl (HDC), (**b**) Mix A: initial, (**c**) Mix A: after 3 h, (**d**) Ctrl (ABPC), (**e**) Mix B: initial, (**f**) Mix B: after 3 h, (**g**) Ctrl (CTX), (**h**) Mix C: initial, (**i**) Mix C: after 3 h, (**j**) Solvent (Normal saline). Detection wavelengths were set at λ = 254 nm for HDC and CTX, and λ = 230 nm for ABPC, respectively. Retention times: HDC = 2.359 min, ABPC = 1.317 min, CTX = 1.717 min.

**Table 1 mps-08-00136-t001:** Summary of samples, additives, diluting solutions, and concentrations.

Drug	Additives	Solvent and Diluent	Concentration
Heparin sodium (Derived from pig intestinal mucosa) (Mochida Pharmaceutical Co., Ltd., Tokyo, 160-0004, Japan)	Sodium chloride benzyl Alcohol	Normal saline	20 U/mL
Hydrocortisone sodium succinate (Pfizer Inc., New York, NY 10017, USA)	Anhydrous disodium phosphate anhydrous monosodium phosphate pH adjusting agent	Normal saline (Dissolved in supplied injection water, 125 mg/mL)	50 mg/mL
Ampicillin sodium (MeijiSeika Pharma Co., Ltd., Tokyo, 104-0031, Japan)	—	Normal saline	250 mg/mL
Cefotaxime sodium (Nichi-Iko Pharmaceutical Co., Ltd., Tokyo, 103-0023, Japan)	—	Normal saline	250 mg/mL

**Table 2 mps-08-00136-t002:** Composition of each mixture.

Mixtures	Composition
Mix A	10 U/mL UFH + 25 mg/mL HDC
Mix B	10 U/mL UFH + 125 mg/mL ABPC
Mix C	10 U/mL UFH + 125 mg/mL CTX
Ctrl (+)	Sodium phosphate corrective injection and calcium gluconate injection (1:1)
Ctrl (−)	water
Ctrl (HDC)	25 mg/mL HDC
Ctrl (ABPC)	125 mg/mL ABPC
Ctrl (CTX)	125 mg/mL CTX

**Table 3 mps-08-00136-t003:** Gradient program and mobile phase composition.

Step	Time (min)	B (%)
1	0	20
2	1	20
3	2	40
4	3	100
5	5	20 (re-equilibration)

Mobile phase A: 0.1% formic acid in water. Mobile phase B: 0.1% formic acid in acetonitrile.

**Table 4 mps-08-00136-t004:** The absorbance of each mixture.

Mixtures	Initial Average (SD)	*p*-Value ^†^	After 3 h Average (SD)	*p*-Value ^†^
Mix A	0.0405 (0.0004)	0.123	0.0419 (0.0007)	0.342
Mix B	0.0419 (0.0011)	0.374	0.0430 (0.0014)	0.700
Mix C	0.0412 (0.0003)	0.225	0.0436 (0.0013)	0.906
Ctrl (+)	1.3761(0.0038)			
Ctrl (−)	0.0438 (0.0038)			

^†^ *p*-value was compared to negative control. The absorbance was measured in triplicate.

**Table 5 mps-08-00136-t005:** pH changes after 3 h of each mixed solution.

Mixtures	Initial Average (SD)	After 3 h Average (SD)	ΔpH	*p*-Value ^‡^
Mix A	7.476 (0.010)	7.539 (0.008)	+0.063	0.0011 *
Mix B	9.142 (0.025)	9.072 (0.000)	−0.070	0.0413 *
Mix C	5.400 (0.001)	5.475 (0.007)	+0.075	0.0031 *

^‡^ *p*-value was compared to the pH at the initial point. The pH was analysed in triplicate. * indicates significant change.

**Table 6 mps-08-00136-t006:** The HPLC/UV peak areas (average value (SD), N = 3) and ratio to control for HDC, ABPC, and CTX.

Mixtures	Peak Area (mAU) of Control	Initial Peak Area (mAU)	Initial Ratio to Control (%)	After 3 h Peak Area (mAU)	After 3 h Ratio to Control (%)	ΔRatio (%)
Mix A	7,192,131 (43,321.7)	6,937,497 (70,065.1)	96.5	6,905,828 (40,071.2)	96.0	−0.5
Mix B	22,759,605 (478,276.1)	24,310,153 (163,183.8)	106.8	24,330,014 (930,607.3)	106.9	+0.1
Mix C	21,091,242 (365,098)	21,901,246 (438,856.4)	96.2	22,298,180 (344,953)	98.0	+1.8

## Data Availability

The data presented in this study are available on request from the corresponding author.

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
