# Peer review of "Compatibility Investigation of a Steroid and Two Antibiotics with Heparin for the Prevention of Catheter Occlusion in Neonatal Intensive Care Units"

_mps, 2025, doi:10.3390/mps8060136_

Round 1

Reviewer 1 Report

Comments and Suggestions for Authors

The authors have done a commendable job, and I find the study quite interesting. My main reservations concern the lack of validation of the analytical method.

  1. In Table 1, the heparin concentration is given as “20 U/ml”; however, in line 92, it appears as “10 U/ml” – please verify.

  1. Lines 138 and 139 – the presentation format of the mobile phase mixtures should be standardized.

  1. In line 167, there is an extra “T” at the beginning of the line.

  1. The brands and origins of the acetonitrile and formic acid should be mentioned in the Methods section.

  1. Although supplementary material is provided, the information presented in the table is overly abbreviated – since the table is a self-explanatory element, the composition of A and B must be clarified within the table itself. The duration of the chromatographic run should also be specified, probably in the main body of the manuscript.

  1. Tables 4 and 5 should present the results of the difference.

  1. The footnote in Table 5 is unnecessary.

  1. In line 200, I believe the authors are referring to their own study, comparing the adopted methodology with those of references 18 and 20. I suggest clarifying more explicitly that the phrase beginning with “Using this approach” refers to the present study.

  1. Line 207: absorbance values are presented in Table 3, not in Table 4 as indicated.

  1. Something seems inconsistent regarding the rationale of Table 4—or perhaps this is a misunderstanding on my part (if so, I apologize and would appreciate clarification). There is an indication of statistical significance in all three rows of the rightmost column, and the text states that the comparison is made with “pH immediately after mixing.” However, comparing the values in the “Immediately after mixing Average (SD)” and “After 3 h Average (SD)” columns suggests that variation was minimal and that statistical difference is highly unlikely (even though p-values below 0.05 indicate statistical significance). Furthermore, in lines 212–217, the authors state that the values remained stable—meaning there was no change after mixing. Therefore, what exactly do the values in the rightmost column of Table 4 refer to?

  1. The issue concerning the lack of calibration curves and the absence of analytical method validation data for the HPLC-UV determinations is more critical than the authors suggest in lines 247–253. One cannot exclude the possibility that the noise level is such that markedly different analyte concentrations would not be detected by the method. Similarly, if peak areas are being compared, it is essential to know whether these areas accurately reflect changes in analyte concentration—something that can only be ensured through analytical validation. I consider this to be the main weakness of the study, and ideally, validation should be performed. Alternatively, if validation is not carried out, the authors should at least clearly acknowledge this limitation; they should cite several references where HPLC-DAD methods were used to compare areas within the same concentration range (at least of the same order of magnitude) without a validated method and calibration curves; and they should include chromatograms.

Author Response

Comments and Suggestions for Authors

The authors have done a commendable job, and I find the study quite interesting. My main reservations concern the lack of validation of the analytical method.

In Table 1, the heparin concentration is given as “20 U/ml”; however, in line 92, it appears as “10 U/ml” – please verify.

Thank you for your comment. You are correct — the correct concentration is 20 U/mL, and the previously stated 10 U/mL was a typographical error. (line 105)

We have corrected this to 20 U/mL in Table 1 and line 105. Accordingly, the same correction was made in the Discussion (line 258). We apologize for the oversight.

The rationale for setting the UFH concentration is as follows:

 In neonatal intensive care units (NICUs), the maximum dose of unfractionated heparin (UFH) is 50 IU/kg/day. For a 5-kg neonate, this corresponds to 250 U/day, or approximately 10.4 U/hour. Assuming an infusion rate of 0.5 mL/hour, a concentration of 20 U/mL appropriately reflects the clinical infusion conditions used in NICUs.

 It should also be noted that when UFH at 20 U/mL is mixed with another drug in a 1:1 ratio at the Y-site, the resulting concentration is diluted to 10 U/mL, which corresponds to the clinically relevant concentration considered in this study.

The rationale for setting the UFH concentration is as follows:

 In neonatal intensive care units (NICUs), the maximum dose of unfractionated heparin (UFH) is 50 IU/kg/day. For a 5-kg neonate, this corresponds to 250 U/day, or approximately 10.4 U/hour. Assuming an infusion rate of 0.5 mL/hour, a concentration of 20 U/mL appropriately reflects the clinical infusion conditions used in NICUs.

 It should also be noted that when UFH at 20 U/mL is mixed with another drug in a 1:1 ratio at the Y-site, the resulting concentration is diluted to 10 U/mL, which corresponds to the clinically relevant concentration considered in this study.

Lines 138 and 139 – the presentation format of the mobile phase mixtures should be standardized.

Thank you for your comment. We have unified the notation of the mobile phase composition by enclosing the ratios in brackets for consistency. The sentence has been revised as follows (line 158-159)

“For separation, the InertSustain C18 PEEK (2.1 mm i.d. × 150 mm, 2 μm, GL Sciences, Tokyo, Japan) was used, with a mobile phase A (formic acid/water [0.1:100, v/v]) and mobile phase B (formic acid/acetonitrile, [0.1:100, v/v]), respectively..”

In line 167, there is an extra “T” at the beginning of the line.

Thank you for pointing this out. The extra “T” at the beginning of the line has been deleted. (line 194)

The brands and origins of the acetonitrile and formic acid should be mentioned in the Methods section.

Thank you for your valuable comment. We have added information on the brand and origin of the acetonitrile and formic acid used in the HPLC mobile phase to the Methods section. The revised text now reads as follows:

Revised text:(Line115-117)

 In the HPLC mobile phase, formic acid (Nacalai Tesque, Kyoto, Japan; ≥98.0% (T), Specially Prepared Reagent), ultrapure water (Organo Corporation, Tokyo, Japan), and acetonitrile (Kanto Chemical Co., Inc., Tokyo, Japan; for HPLC, >99.9% (GC)) were used.

Although supplementary material is provided, the information presented in the table is overly abbreviated – since the table is a self-explanatory element, the composition of A and B must be clarified within the table itself. The duration of the chromatographic run should also be specified, probably in the main body of the manuscript.

Thank you for your valuable suggestion. In response to another reviewer’s comment, the supplementary table has been incorporated into the main manuscript as Table 3. The compositions of mobile phases A and B are now clearly indicated within the table for clarity. In addition, the total chromatographic run time (5 min) has been specified in Section 2.6 of the main text (lines 159–161).

Tables 4 and 5 should present the results of the difference.

Thank you for your helpful comment. We have added the difference (Δ) values to Tables 4 and 5 to clearly present the changes in pH and the ratios to control after 3 h. The column titles were revised accordingly (ΔpH and ΔRatio (%))(Tables 5 and 6).

The footnote in Table 5 is unnecessary.

Thank you for your comment. The footnote in Table has been deleted(Table 6).

In line 200, I believe the authors are referring to their own study, comparing the adopted methodology with those of references 18 and 20. I suggest clarifying more explicitly that the phrase beginning with “Using this approach” refers to the present study.

Thank you for your suggestion. To clarify that “this approach” refers to the method used in the present study, we revised the sentence as follows:

In the present study, we applied this approach and measured the absorbance at 685 nm; water exhibited the lowest absorbance.

  (lines 233-235)

Line 207: absorbance values are presented in Table 3, not in Table 4 as indicated.

Thank you for pointing this out. Following the revision requested in another comment, the HPLC gradient information originally presented in the Supplementary Materials has been moved into the main text as Table 3. Consequently, the absorbance table has been renumbered from Table 3 to Table 4. The reference in line 207 has been corrected accordingly to Table 4 (lines 240).

Something seems inconsistent regarding the rationale of Table 4—or perhaps this is a misunderstanding on my part (if so, I apologize and would appreciate clarification). There is an indication of statistical significance in all three rows of the rightmost column, and the text states that the comparison is made with “pH immediately after mixing.” However, comparing the values in the “Immediately after mixing Average (SD)” and “After 3 h Average (SD)” columns suggests that variation was minimal and that statistical difference is highly unlikely (even though p-values below 0.05 indicate statistical significance). Furthermore, in lines 212–217, the authors state that the values remained stable—meaning there was no change after mixing. Therefore, what exactly do the values in the rightmost column of Table 4 refer to?

Thank you very much for your careful reading and for pointing out this issue.

For Mix A (10 U/mL UFH + 25 mg/mL HDC), the pH increased by +0.063 after 3 hours. The statistical analysis indicating a significant difference in Table 4 is correct. Therefore, the previous expression “the pH of HDC after mixing remained within a stable range” was inappropriate.

However, since HDC is known to remain stable within the pH range of 7.0–8.0, an increase of +0.063 is considered to be within the range that does not affect its stability. To clarify this point, we have revised the text as follows:

Revised sentence:(Line245-247)

In this study, the pH of HDC increased by +0.063 after mixing. However, since HDC is known to remain stable within the pH range of 7.0–8.0, this slight increase is within the range that does not affect its stability.

The issue concerning the lack of calibration curves and the absence of analytical method validation data for the HPLC-UV determinations is more critical than the authors suggest in lines 247–253. One cannot exclude the possibility that the noise level is such that markedly different analyte concentrations would not be detected by the method. Similarly, if peak areas are being compared, it is essential to know whether these areas accurately reflect changes in analyte concentration—something that can only be ensured through analytical validation. I consider this to be the main weakness of the study, and ideally, validation should be performed. Alternatively, if validation is not carried out, the authors should at least clearly acknowledge this limitation; they should cite several references where HPLC-DAD methods were used to compare areas within the same concentration range (at least of the same order of magnitude) without a validated method and calibration curves; and they should include chromatograms.

Thank you very much for your insightful and constructive comments. As you correctly pointed out, this study lacks calibration curves and analytical method validation data for the HPLC-DAD analysis, and therefore the quantitative reliability of the measurements is limited.

In response, we have revised the Abstract to clearly describe the HPLC analysis as semi-quantitative. In addition, in the Discussion, we explicitly acknowledge this as a limitation, stating that the results represent relative rather than absolute changes in analyte concentrations.

Because published data on the compatibility of injectable admixtures are limited, many previous studies have relied primarily on physicochemical evaluations, such as visual inspection, turbidity, and absorbance measurements, while chemical stability assessments have often been restricted to pH measurements (Refs. 18, 20, 21, 23).

Furthermore, in practical clinical settings, even the preparation of infusion mixtures can involve a certain degree of variability. For example, a previous study on volume accuracy using medical syringes reported a maximum error of 8.84% (JJOMT, 63:31–35, 2015). In addition, errors related to preparation procedures and residual drug volumes in syringes or catheters are generally acceptable within a broader range than analytical quality control levels.

Although our study does not provide absolute quantitative evaluation, we believe that presenting the data as semi-quantitative results is appropriate for the purpose of this work—namely, to determine whether HDC, ABPC, and CTX can be safely administered via the side port during continuous UFH infusion in NICU practice.

[abstract]

In semi-quantitative HPLC analysis, (Line21)

[Discussion]

Line290-295

Reviewer 2 Report

Comments and Suggestions for Authors

Dear authors,

presented paper has practical value and has value for health practice in NICU.

I have suggestions as follows:

introduction  should have more parts of concomitant application of UFH with HDC, ABPC and CTX. Pleas do explain what is dosage that is usually applied. Does HDC, ABPC and CTX could be applied together, or is there always more than 3 h apart.

2.2. Mixing of drugs for preparation of test samples -----reasoning for selected concentrations. Is it possible to have higher UFH and lower drug concentration. Is 10 U/ml maximal concentration.

Is it possible that UFH could be mixed with HDC, ABPC, CTX at the same time or two of this drugs (not only one as proposed in the investigation).

Absorbance measurement-----is this used for turbidimetry, was there an end criteria to compared level of solution cloudiness. 

was the temperature for pH measurement constant?

Measurement of changes in HDC, ABPC, and CTX content-----could UFH content be changed or affected, please discusse this

good luck

Author Response

Comments and Suggestions for Authors
Dear authors,
presented paper has practical value and has value for health practice in NICU.
I have suggestions as follows:

introduction  should have more parts of concomitant application of UFH with HDC, ABPC and CTX. Pleas do explain what is dosage that is usually applied. Does HDC, ABPC and CTX could be applied together, or is there always more than 3 h apart.

Thank you for your helpful comment. To clarify the clinical dosing rationale and administration practices, we have added detailed descriptions in the Introduction(①~③). In neonatal intensive care, unfractionated heparin (UFH) is administered at a maximum dose of 50 IU/kg/day. For a 5-kg neonate, this corresponds to 250 U/day, or approximately 10.4 U/hour. Assuming an infusion rate of 0.5 mL/hour, we determined that a 20 U/mL concentration appropriately reflects the clinical infusion conditions used in NICUs.
When UFH prepared at 20 U/mL is mixed 1:1 with other drugs on the Line, the concentration is diluted to half, 10 U/mL.
Regarding the other study drugs, hydrocortisone sodium succinate (HDC) is typically administered at 3.75–5 mg/kg/day in extremely preterm infants on mechanical ventilation, as reported in Reference 8. Ampicillin sodium (ABPC) and cefotaxime sodium (CTX) are administered at 100–400 mg/kg/day and 50–150 mg/kg/day, respectively, according to product information. As the dosing concentrations of these drugs are not specified, we performed testing at their maximum soluble concentrations.

Although co-administration of HDC, ABPC, and CTX with UFH may occur in practice, these agents are usually administered as rapid intravenous injections rather than by continuous infusion, unlike UFH. Therefore, by avoiding simultaneous administration, direct contact between these drugs can be prevented. For this reason, combination tests among HDC, ABPC, and CTX themselves were not conducted in this study.

[I added a postscript in introduction]
â‘ In the NICU, UFH is typically diluted to at most 20 U/mL to meet neonatal dosing requirements (maximum 50 IU/kg/day in infants weighing 0.5–5 kg) and is administered at infusion rates of 0.5–1 mL/h [14–16], indicating the need to evaluate compatibility under these clinically relevant conditions.(Line63-67)
â‘¡This study aimed to evaluate the physical and chemical compatibility of HDC, ABPC, and CTX—three drugs frequently co-administered in NICUs—when in contact with 20 U/mL UFH under conditions simulating Y-site administration. The concentrations of each drug were set at their maximum clinically used concentrations. Because these antibiotics can be administered intermittently to avoid direct admixture, compatibility among HDC, ABPC, and CTX themselves was not investigated.(Line77-82)
â‘¢ In contrast, unfractionated heparin (UFH) is a highly sulfated polysaccharide with a broad molecular weight distribution ranging from approximately 5,000 to 20,000 Da [24], making it unsuitable for separation or quantification by conventional HPLC-UV methods. Therefore, in this study, quantitative analysis was limited to the three low molecular weight compounds, whereas UFH compatibility was assessed primarily based on physical parameters such as turbidity, precipitation, and pH change.
Therefore, using this combined analytical approach, we investigated the compatibility of HDC, ABPC, and CTX with UFH under conditions simulating continuous infusion and Y-site administration in NICU clinical practice.(Line91-100)

2.2. Mixing of drugs for preparation of test samples -----reasoning for selected concentrations. Is it possible to have higher UFH and lower drug concentration. Is 10 U/ml maximal concentration.

The UFH concentration of 20 U/mL is considered the maximum level used in NICU clinical practice, because, as mentioned in the Introduction, it was determined based on a 5-kg neonate receiving the maximum daily dose of 50 IU/kg/day. In actual clinical settings, most neonates weigh less than 5 kg and receive intravenous therapy accordingly.

Is it possible that UFH could be mixed with HDC, ABPC, CTX at the same time or two of this drugs (not only one as proposed in the investigation).

Thank you for your question. As mentioned in our response to the first comment, HDC, ABPC, and CTX may be administered in combination with UFH in NICU clinical practice. However, unlike UFH, which is administered by continuous infusion, these drugs are typically given by rapid intravenous injection. Therefore, avoiding simultaneous administration effectively prevents direct contact between these drugs. For this reason, combinations among HDC, ABPC, and CTX themselves were not evaluated in this study.

Absorbance measurement-----is this used for turbidimetry, was there an end criteria to compared level of solution cloudiness. 

Thank you for your comment. As you mentioned, turbidity measurement is one of the methods commonly used to evaluate physical compatibility in admixture studies, as described in several previous reports (References 21, 23, 32, and 36).
 Turbidimetry measures scattered light, whereas absorbance measurement determines transmitted light, and therefore these are different approaches for assessing solution clarity.
In the Japanese Pharmacopoeia, visual inspection (appearance test) is the standard method used to confirm the absence of precipitation or suspension in injectable preparations. Depending on the purpose of the evaluation, turbidity or absorbance measurements may also be used as quantitative methods. However, there is no official criterion or regulatory standard specifying which technique should be applied.
In this study, we followed the approach described by Kenneally et al. [18] and Dobson et al. [20], using absorbance measurement to assess solution clarity. The samples showed comparable transmittance to normal saline, which is known to be clear and free of particulates, indicating that no turbidity or precipitation occurred during the test period.

was the temperature for pH measurement constant?

The pH measurements were conducted at a constant room temperature. All samples were equilibrated to room temperature prior to measurement to ensure that the solution temperature remained constant during the experiment.

Measurement of changes in HDC, ABPC, and CTX content-----could UFH content be changed or affected, please discusse this

Thank you for your valuable comment. The possibility that the UFH content may be affected cannot be completely excluded. As described in the revised Introduction (see paragraph â‘¢), UFH has a broad molecular weight distribution (approximately 5,000–20,000 Da), which makes it difficult to separate and quantify by conventional HPLC methods. Therefore, UFH content was not measured in this study.
As an alternative approach, anticoagulant activity assays could be used to evaluate potential changes in UFH activity in admixtures. However, in clinical practice, the anticoagulant effect of UFH is routinely monitored by blood tests, and the infusion rate can be adjusted accordingly. For this reason, we considered that a detailed quantitative analysis of UFH content was not essential in this compatibility study compared with HDC, ABPC, and CTX.

good luck

Reviewer 3 Report

Comments and Suggestions for Authors

In this study, the authors investigated the compatibility of hydrocortisone sodium succinate, ampicillin sodium, and cefotaxime sodium when co-administered with unfractionated heparin. To confirm the compatibility of these mixtures, visual inspection was performed to check for precipitation or turbidity. Additionally, pH measurements were conducted to assess the pH level of the solutions. Finally, comparative peak areas were compared to evaluate the stability of the mixtures over time. The research design is sound. The introduction provides adequate background information. However, I would like to suggest some improvements to enhance the content for the readers.

Minor comments:

  1. Line 100: correct the (ctrl (+)) and Line 112: (Ctrl [-)].

Major comments:

  1. The purpose of this study, in lines 85~87, is missing the main coposition material, unfractionated heparin. Correct the section.
  2. The mixture compositions were compared at the initial and 3h time point. It would be Ideal to correct the expression “Immediately after mixing” to Initial.
  3. Visual observation was used to evaluate the crystal precipitation; for accurate observation, microscopic analysis is suggested.
  4. UV absorbance and pH Data should be presented for all three controls: Ctrl (HDC), Ctrl (ABPC), and Ctrl (CTX).
  5. It is inefficient to provide gradient information in the supplementary info; rather can be explained side by side with the text.

Author Response

Comments and Suggestions for Authors
In this study, the authors investigated the compatibility of hydrocortisone sodium succinate, ampicillin sodium, and cefotaxime sodium when co-administered with unfractionated heparin. To confirm the compatibility of these mixtures, visual inspection was performed to check for precipitation or turbidity. Additionally, pH measurements were conducted to assess the pH level of the solutions. Finally, comparative peak areas were compared to evaluate the stability of the mixtures over time. The research design is sound. The introduction provides adequate background information. However, I would like to suggest some improvements to enhance the content for the readers.

Minor comments:

Line 100: correct the (ctrl (+)) and Line 112: (Ctrl [-)].

We have corrected the notations as suggested.
(ctrl (+)) → (Ctrl [+]) (Line 113)
(Ctrl [-)]) → (Ctrl [-]) (Line 128)

Major comments:

The purpose of this study, in lines 85~87, is missing the main coposition material, unfractionated heparin. Correct the section.

Thank you for your valuable comment. We have revised the Introduction section based on the reviewers’ suggestions, and the purpose of the study now clearly includes unfractionated heparin (UFH). Specifically, the following sentence has been added:
“Therefore, using this combined analytical approach, we investigated the compatibility of HDC, ABPC, and CTX with UFH under conditions simulating continuous infusion and Y-site administration in NICU clinical practice.” (Lines 99–100)
This revision clarifies that UFH is one of the main components evaluated for compatibility in this study.

The mixture compositions were compared at the initial and 3h time point. It would be Ideal to correct the expression “Immediately after mixing” to Initial.

Thank you for your suggestion. We have carefully revised the expressions throughout the manuscript to improve consistency and clarity regarding the sampling time points. Specifically, the following changes were made:
Lines 18, 124: The adverbial expressions “immediately and after 3 h” were retained, as they describe the sampling procedure.
Lines 144, 150, 163 (Methods): Changed “measured immediately after mixing” to “measured at the initial and 3 h time points.”
Lines 175, 188, 220, 220, 269, 285 (Results): Changed “Immediately after mixing and after 3 h” to “at the initial and 3 h time points.”
Line 178: Changed “became turbid immediately after mixing” to “became turbid at the initial point.”
Figure 1: All occurrences of “immediately after mixing” were changed to “initial.”
Tables 4-6: Changed “Immediately after mixing Average (SD)” to “Initial Average (SD).”
These revisions align the terminology across the manuscript and clearly distinguish between the initial and 3 h measurement time points.

Visual observation was used to evaluate the crystal precipitation; for accurate observation, microscopic analysis is suggested.

We sincerely appreciate the reviewer’s insightful suggestion. We fully agree that microscopic analysis can provide a more detailed assessment by enabling detection of smaller, sub-visible particles. In this study, however, visual inspection—that is, observation with the naked eye—was selected because it is a widely accepted and pharmacopeially standardized method for detecting visible incompatibilities such as precipitation, turbidity, and color change in parenteral preparations.
According to the Japanese Pharmacopoeia (JP, Test for Foreign Insoluble Matter in Injections, 6.06), visual inspection of injections is conducted under white light (2000–3750 lx) against both black and white backgrounds for approximately five seconds, and the preparation must be free of readily detectable insoluble matter. This standard supports that visual inspection represents an appropriate and validated method for evaluating visible physical changes.
Consistent with numerous previous compatibility studies (Refs. 18, 20, 21, 23, 38, 39), we employed visual inspection as the initial screening method, complemented by UV absorbance measurements to quantitatively detect subtle or sub-visible changes. We believe that this combined approach—macroscopic visual inspection together with quantitative spectrophotometric analysis—provides sufficient sensitivity and reliability to support the conclusions of this study regarding the absence of physicochemical incompatibility.
We are grateful for the reviewer’s comment and will take this valuable advice into account when designing future studies that may incorporate microscopic or particle-counting techniques to further refine the analytical accuracy.

UV absorbance and pH Data should be presented for all three controls: Ctrl (HDC), Ctrl (ABPC), and Ctrl (CTX).

We sincerely appreciate the reviewer’s valuable and constructive comment. We fully agree that presenting individual control data can provide additional clarity regarding the intrinsic absorbance and pH characteristics of each drug.
In this study, however, the negative control for UV absorbance measurement was set as normal saline (0.9% sodium chloride injection). The saline used complies with the Japanese Pharmacopoeia (JP) specifications for Foreign Insoluble Matter <6.06> (Method 1) and Insoluble Particulate Matter <6.07>, which ensures the absence of visible and subvisible particulates. Therefore, when the absorbance of each test mixture was comparable to that of the saline control, it was considered that no precipitation or turbidity had occurred.
We acknowledge that some previous compatibility studies have included individual controls for each drug (e.g., Refs. 21 and 23), whereas others have used only sterile water or saline as a single negative control (Refs. 18, 20, 38, 39). Following this latter approach, our study employed saline as a universal control verified to be free from precipitation. The absorbance values of all tested mixtures showed no deviation from that of the saline control, supporting the absence of insoluble particles or precipitation under the study conditions.
We also agree with the reviewer’s opinion that conducting individual control experiments for each drug would help confirm their intrinsic spectral and pH stability and thereby further strengthen the interpretation of the results. In future studies, particularly when expanding the compatibility evaluation to additional drugs, if measurable differences are observed compared with the saline control, we plan to perform individual control testing for each drug to further validate the reliability of the Y-site compatibility assessment.
Regarding pH measurement, it was not used as a direct determinant of physical compatibility but rather as a supportive parameter to monitor potential physicochemical changes over time, as also described in prior studies (Refs. 18, 20, 38, 39). Thus, the pH data were evaluated as time-dependent deviations from baseline (0 h), not as absolute comparisons among controls.
Although published compatibility data are still limited—especially concerning neonatal intensive care unit (NICU) practice—our findings, together with these methodological precedents, support the conclusion that continuous infusion of UFH is physically compatible with HDC, ABPC, and CTX under the tested conditions. We will take the reviewer’s valuable suggestion into account when refining our experimental design in future studies.
Planned Revisions in Manuscript
Methods (Control section – revised sentence):(Line141-143)
 “Normal saline (0.9% NaCl injection), compliant with JP <6.06> and <6.07>, was used as the negative control and blank for spectrophotometric analysis. Mixtures exhibiting absorbance comparable to saline were considered free of precipitation or turbidity.”
Results (clarified statement):(Line188-195)
3.2. Absorbance
The UV absorbance of each mixture, both at the initial and 3 h time points, showed no significant difference compared with the saline control (Table 4). This indi-cates that no significant spectral changes occurred, suggesting the absence of visible or subvisible particulate formation.
3.3. pH change
The pH of mixes A and C increased after 3 h, whereas that of Mix B decreased (Table 5). Overall, no significant deviation from baseline pH was observed, further supporting the stability of the mixtures during the observation period.

It is inefficient to provide gradient information in the supplementary info; rather can be explained side by side with the text.

We sincerely appreciate the reviewer’s valuable suggestion. In response, we have moved the HPLC gradient program from the Supplementary Information into the Methods section (Section 2.6) and described it alongside the details of the column and mobile phases to improve readability and reproducibility. Specifically, the sentence “The gradient program and mobile phase compositions are presented in Supplementary Table 1.” has been replaced with a concise in-line description of the gradient (0.00–1.00 min, 20% B; 1.00–2.00 min, 20%→40% B; 2.00–5.00 min, 100% B; post-run re-equilibration to 20% B).
Furthermore, Supplementary Table 1 has been incorporated into the main text as Table 3 to present the gradient program directly within the manuscript. (Line159-160)

Round 2

Reviewer 1 Report

Comments and Suggestions for Authors

The authors have satisfactorily addressed most of the comments raised in the previous review round, and the revised version shows clear improvements in clarity and scientific rigor.

However, regarding the issue of analytical method validation, I would like to reiterate that, even if the HPLC analysis is now described as semi-quantitative, it remains essential to provide at least minimal evidence of reliability. Specifically, the inclusion of representative chromatograms would considerably strengthen the credibility of the analytical results.

If the chromatographic analysis is presented as an added value of the study, then there should be some demonstration—however limited—that the results obtained are consistent and interpretable. The addition of chromatogram images, even without full validation data, would help readers assess the selectivity and general adequacy of the method used.

Author Response

Comments and Suggestions for Authors
The authors have satisfactorily addressed most of the comments raised in the previous review round, and the revised version shows clear improvements in clarity and scientific rigor.

However, regarding the issue of analytical method validation, I would like to reiterate that, even if the HPLC analysis is now described as semi-quantitative, it remains essential to provide at least minimal evidence of reliability. Specifically, the inclusion of representative chromatograms would considerably strengthen the credibility of the analytical results.

If the chromatographic analysis is presented as an added value of the study, then there should be some demonstration—however limited—that the results obtained are consistent and interpretable. The addition of chromatogram images, even without full validation data, would help readers assess the selectivity and general adequacy of the method used.

We appreciate this valuable suggestion.
In response, representative chromatograms for each mixture (Mix A–C) at the initial and 3 h time points have been added as Figure 2 in the Results section (Section 3.4). The figure clearly shows single, well-defined peaks at the expected retention times without interference from coexisting compounds, thereby confirming the selectivity and reliability of the HPLC/UV analysis. Detection wavelengths and retention times are described in the figure legend (λ = 254 nm for HDC and CTX, λ = 230 nm for ABPC; retention times: HDC = 2.359 min, ABPC = 1.317 min, CTX = 1.717 min).

In addition, to further improve the transparency and reliability of the data, Table 6 was revised. The previous version showed only the ratios to control, whereas the revised table now includes the average ± SD of the peak areas for both the control and the mixtures at each time point. This allows a more comprehensive evaluation of quantitative changes in HDC, ABPC, and CTX.

These revisions substantially strengthen the analytical validity and reproducibility of the presented results.

Reviewer 3 Report

Comments and Suggestions for Authors

Dear Authors,

Thanks for your effort in revising the manuscript. 

It’s great to see the additional wording in the methods and results sections.

Author Response

Comments and Suggestions for Authors
Dear Authors,

Thanks for your effort in revising the manuscript. 
It’s great to see the additional wording in the methods and results sections.

Thank you for your positive feedback.
We appreciate your recognition of the improvements, and we believe that these revisions have resulted in a higher-quality manuscript.
